# Understanding the Impact of Initial COVID-19 Restrictions on Physical Activity, Wellbeing and Quality of Life in Shielding Adults with End-Stage Renal Disease in the United Kingdom Dialysing at Home versus In-Centre and Their Experiences with Telemedicine

**DOI:** 10.3390/ijerph18063144

**Published:** 2021-03-18

**Authors:** Joe Antoun, Daniel J. Brown, Daniel J. W. Jones, Nicholas C. Sangala, Robert J. Lewis, Anthony I. Shepherd, Melitta A. McNarry, Kelly A. Mackintosh, Laura Mason, Jo Corbett, Zoe L. Saynor

**Affiliations:** 1School of Sport, Health and Exercise Sciences, Faculty of Science and Health, University of Portsmouth, Portsmouth PO1 2UP, UK; Joe.antoun@port.ac.uk (J.A.); daniel.brown@port.ac.uk (D.J.B.); ant.shepherd@port.ac.uk (A.I.S.); Jo.Corbett@port.ac.uk (J.C.); 2Academic Department of Renal Medicine, Wessex Kidney Centre, Portsmouth Hospitals NHS Trust, Portsmouth PO6 3LY, UK; Nicholas.sangala@porthosp.nhs.uk (N.C.S.); Robert.Lewis@porthosp.nhs.uk (R.J.L.); 3School of Psychology and Clinical Language Sciences, University of Reading, Reading RG6 6AH, UK; dan.jones@reading.ac.uk; 4Applied Sports, Technology, Exercise and Medicine Research Centre (A-STEM), School of Sport and Exercise Sciences, Swansea University, Swansea SA2 8PP, UK; M.Mcnarry@Swansea.ac.uk (M.A.M.); K.Mackintosh@Swansea.ac.uk (K.A.M.); l.mason@swansea.ac.uk (L.M.)

**Keywords:** quality of life, health behaviour, doctor–patient communication, experience of illness and disease, exercise, user experiences, nephrology, internet technology, wellbeing

## Abstract

Early in the coronavirus-2019 (COVID-19) containment strategy, people with end-stage renal disease (ESRD) were identified as extremely clinically vulnerable and subsequently asked to ‘shield’ at home where possible. The aim of this study was to investigate how these restrictions and the transition to an increased reliance on telemedicine within clinical care of people living with kidney disease impacted the physical activity (PA), wellbeing and quality of life (QoL) of adults dialysing at home (HHD) or receiving in-centre haemodialysis (ICHD) in the UK. Individual semistructured telephone interviews were conducted with adults receiving HHD (*n* = 10) or ICHD (*n* = 10), were transcribed verbatim and, subsequently, thematically analysed. As result of the COVID-19 restrictions, PA, wellbeing and QoL of people with ESRD were found to have been hindered. However, widespread support for the continued use of telemedicine was strongly advocated and promoted independence and satisfaction in patient care. These findings highlight the need for more proactive care of people with ESRD if asked to shield again, as well as increased awareness of safe and appropriate PA resources to help with home-based PA and emotional wellbeing.

## 1. Introduction

Early in the coronavirus 2019 (COVID-19) global pandemic, people with some pre-existing medical conditions were identified as having an increased risk. Since chronic kidney disease (CKD) is associated with a greater risk of both inpatient and outpatient pneumonia [1] and a 14–16 times higher pneumonia-related mortality rate than the general population [2], it was unsurprising when early reports from a haemodialysis (HD) facility in Wuhan reported a much higher infection rate in people with end-stage renal disease (ESRD) [3]. Indeed, a meta-analysis of pooled data also revealed a significant association between CKD and more severe COVID-19 infection [4] and a much greater mortality rate than the general population. Consequently, in the UK people with ESRD receiving dialysis were classified as ‘extremely clinically vulnerable’ in April 2020 and asked to shield at home where possible [4].

People with ESRD were now experiencing a change to much stricter regulations. Although aimed at limiting COVID-19 transmission, these new restrictions reduced social contact and physical activity (PA), potentially leading to negative health implications [5,6]. However, unlike other clinical groups, fully shielding people with ESRD from COVID-19 was not possible, since they require regular, essential renal replacement therapy (dialysis) and clinical monitoring [7]. Travelling to a dialysis facility for lengthy treatments in close proximity to dialysis staff and other people receiving dialysis may also engender additional COVID-19 risks, which is of particular concern given the reduced immune function of people receiving dialysis [8]. Nonetheless, reducing the frequency of dialysis sessions in an attempt to mitigate the risk of COVID-19 transmission would greatly increase the risk of cardiovascular events, or death [9].

General population data and reports from early in the COVID-19 pandemic highlighted that shielding can have a detrimental impact on the mental health and wellbeing of older people [6,10], as well as healthy younger adults not shielding but living under restrictions to human movement [11]. PA has been identified as a key modifying factor of mental health, with data from a multinational study reporting that a negative change in exercise behaviour during the initial COVID-19 restrictions was associated with poorer mental health and wellbeing [12]. This is of concern, since poor quality of life (QoL) and wellbeing were already known to characterise people with CKD and ESRD before COVID-19 [13,14,15], particularly those receiving dialysis [16,17]. Furthermore, QoL and wellbeing are reported to be negatively influenced when dialysis is in the form of in-centre haemodialysis (ICHD) [18,19,20]. Indeed, maintenance ICHD is also associated with reduced habitual PA, even when patients are relatively healthy [21,22,23], likely contributing to their typically diminished physical function [24]. Any additional impact of shielding from COVID-19 on these factors in people receiving dialysis remains to be elucidated.

An additional consequence of the COVID-19 pandemic was a rapid change in the way clinical care was delivered. Specifically, face-to-face contact between clinicians and people living with kidney disease was abruptly reduced and replaced by an increased reliance on telemedicine and remote clinics [25,26,27]. This change in practice is likely to persist following the COVID-19 pandemic, at least to some extent, in accordance with the 2019 NHS Long Term Plan [27,28]. It was therefore important to determine the experiences of people with ESRD in receipt of this altered care delivery.

The present study therefore aimed to explore the experiences of the initial COVID-19 restrictions on PA, wellbeing and QoL in shielding adults with ESRD in the UK receiving ICHD or HHD, as well as their experiences of telemedicine.

## 2. Materials and Methods

This study consisted of individual semistructured interviews conducted over the telephone between 10 June and 10 July 2020. Semistructured interviews provide a rich, personal narrative of participants’ experiences and rely upon the rapport and relationship between researcher and participant. This time period was chosen to correspond with the initial COVID-19 containment strategies concerning human movement in the UK and following the reclassification of people with CKD and ESRD as ‘extremely clinically vulnerable’. The study was designed and conducted in line with Guba and Lincoln’s criteria for rigour in qualitative research [29] and reported using the COnsolidated criteria for REporting Qualitative research (COREQ) checklist [30].

### 2.1. Participants

Adults with ESRD currently under the care of the Wessex Kidney Centre and receiving either ICHD (*n* = 10; 61.2 ± 8.9 years; 8 males; time on dialysis: 44 ± 58 months) or HHD (*n =* 10; 65.4 ± 11.8 years; 6 males; time on dialysis, 61 ± 46 months) were invited by a member of their primary clinical team or the principle researcher (JA) to participate. Fully informed verbal consent was obtained prior to participation. Inclusion criteria consisted of being ≥18 years of age, willing and able to provide consent and being free from any psychiatric or neurological diagnosis at the time of interview. Recruitment continued until data saturation was reached (i.e., little new data was appearing within interviews and concepts were well developed [31]. Several additional interviews were conducted to confirm data saturation, totalling 10 in each group.

### 2.2. Dialysis Management

People receiving ICHD must attend three sessions per week, which typically last 3–5 h depending on the time required to achieve adequate waste removal and fluid removal. Most of these individuals also require patient transport services to assist with travel to and from their dialysis unit. The Wessex Kidney Centre HHD programme exclusively uses the NxStage^®^ System 1 or 1S dialysis machines (NxStage Medical Inc, Merrimack St. Lawrence, MA, USA), with dialysis prescriptions varying between short daily dialysis to alternate day nocturnal regimens, utilising dialysate volumes ranging from 20 to 60 L. Dialysis prescriptions are adjusted to achieve a standardised weekly adequacy (2.4 Kt∙V-1), with flexibility given to individuals with a significant residual urine volume.

### 2.3. Telemedicine

Before the COVID-19 pandemic, people with ESRD attended routine quarterly face-to-face clinics with their renal consultant. For those on ICHD, this usually involved attending the dialysis centre and conducting a consultation at the bedside. As a result of COVID-19 restrictions, it was necessary to undertake these routine reviews by telephone rather than face-to-face. For people with additional comorbidities (e.g., diabetes, cardiac or respiratory conditions), additional telephone clinics were also required with other specialties. For people receiving HHD (but not those on ICHD), the Wessex Kidney Centre HHD introduced a digital remote monitoring platform (MyRenalCare^®^-https://www.myrenalcare.com/) before the COVID-19 restrictions were implemented. This system was developed by Ardia Digital Health Ltd.^®^ in collaboration with the Wessex Kidney Centre. It is a purpose-built, integrated telehealth platform for people with ESRD and their clinicians, which aims to help people living with kidney disease take ownership of recording clinical and technical information about their HHD sessions and patient-reported outcome measures. This was continued during the COVID-19 restrictions.

### 2.4. Data Collection

Individual semistructured telephone interviews were conducted by JA. Given that this population were still largely isolating at home due to the initial UK COVID-19 restrictions, face-to-face interviews were not permitted. Through initial Patient and Public Involvement with people receiving dialysis and their clinicians, as well as previous attempts to use alternative technology for research with this cohort, a telephone call was deemed more feasible than video conferencing software. Participants were asked to be at home and ensure they were ‘comfortable’ with their surroundings. For people receiving ICHD, interviews were conducted on a non-dialysis day. Since most people receiving HHD dialyse six days per week, conducting interviews on non-dialysis days was not always possible and thus they were conducted at a different time from their dialysis session, thereby minimising any acute impact of the dialysis session.

An interview guide comprising 15 open-ended questions was developed through a review of the literature and informal discussions with potential participants and the research design team. Specifically, the interview guide sought to explore aspects of the participants’ PA, wellbeing and QoL during this period, as well as their experiences of the increased use of telemedicine within their clinical care. Participants were informed that the researcher conducting the interviews was not part of their clinical care team and was interested in understanding the impact of the COVID-19 restrictions on all aspects of their life, for example: “How are you feeling about the current situation and the changes caused by the COVID-19 outbreak?” and “Please can you tell me about your current experiences with medical appointments?” During the interviews the topic guide was used flexibly, with prompts and probes used to encourage participants to expand and elaborate on topics when needed and to include any additional information they felt necessary. The interviews ranged from 22 to 75 min (median length: 30 min) and were concluded when each participant felt that they had nothing further to add. The interviews were audio recorded and then transcribed verbatim. All audio recordings/transcripts were only available to members of the research team and all transcripts were anonymised immediately.

### 2.5. Data Analysis

Transcripts were analysed using NVivo 12 software package (QSR International, Doncaster, Australia) and coding and thematic analysis undertaken using a systematic approach [32]. An abductive approach was taken to combine codes to generate families of ideas or similar over-arching themes. First, a code manual was developed to organise similar ideas and relate texts to assist with the interpretation of the data. The second stage involved summarising the data individually and identifying any initial themes (conducted by J.A., D.J.B. and Z.L.S.). Third, the initial themes were revisited, and codes applied, before being regrouped into more definite groups. Fourth, the final codes and themes were corroborated (conducted by J.A. and D.J.B.). Finally, the results were discussed with several other researchers within the team so that internal thinking processes were made explicit, ideas clarified, and new insights obtained.

## 3. Results

Twenty adults with ESRD (10 receiving HHD and 10 receiving ICHD) were enrolled in this study. Throughout the interviews in this study, it became clear that people expressed the impact of the COVID-19 restrictions through how aspects of their lives had changed or indeed remained similar to before the pandemic. Within this, the dimensions of PA, wellbeing and QoL, and medical care were discussed, and several key themes identified (Figure 1).

### 3.1. HHD Group

#### 3.1.1. Dimension: PA

##### Maintained PA

Some of the participants described how their levels of PA have either remained the same or even improved. Indeed, one individual stated that their PA did not change, “because we have largely gone about our normal business and near here there are walks around here that you can do without meeting anyone anyway”. Similarly, another stated: “I go for quite long walks with the dog. Again, it’s only in the fields and stuff, I don’t go to any built-up areas, but I am following the guidelines to protect myself, but I am also kind of stretching them a bit to suit myself”.

##### Changes in PA

With regard to PA, participants described two changes as a result of the COVID-19 restrictions, with the first being changes in frequency of PA, where individuals described a notable change in the amount of time spent being physical active compared to before the COVID-19 restrictions. One participant said: “Oh, well yes they [PA levels] have gone down quite a lot obviously”. On the contrary, one participant described an increase in the amount of PA they were undertaking, due to the impact of working from home and the extra time this gave, saying: “I think, they have actually gone up, randomly. I really took the opportunity because I enjoy my running, so I have really taken the opportunity to get back into that and yes, we have been cycling regularly”, which suggests not all individuals were strictly shielding.

The second change was in the modality of PA, which saw individuals changing their approach to PA as a result of the restrictions, including gardening or indoor alternative to exercise. For example, one participant said “Yes, so I started doing P.E. [Physical Education] with Joe Wicks and I started doing that Monday to Friday”. Another participant said: “We are lucky we have a garden, so we have been doing gardening. We have had plants delivered and made use of the garden and working in the garden”.

#### 3.1.2. Dimension: Wellbeing and QoL

##### Aspects Maintained

Participants in the HHD group described four areas which they felt had remained constant from prior to the COVID-19 pandemic and associated restrictions in relation to their wellbeing and QoL. The first of these areas related to the participants’ employment, with a number of individuals stating that they felt the support received from colleagues had remained consistent. For example, one participant said, “one of the assistant managers phoned me later and then I felt supported”. A second area described was the stability in their psychological wellbeing, which had remained constant throughout COVID-19. For example, one participant said: “Predominantly over the last three years I have been having counselling [to support my wellbeing] and I think yes, COVID-19 was in there [so my psychological wellbeing has remained constant]”. Thirdly, participants described their renal treatment and how their exposure to a chronic condition prepared them for the social requirements of COVID-19, with one participant stating: “We had to alter our lives so dramatically because of dialysis, which is a daily event … in a way, we had a prelude to losing freedom”. Finally, participants described their experience as having no change, with many expressing the lack of effect as a result of the COVID-19 restrictions. For example, when asked what had changed, one participant stated: “Well not a lot really because we didn’t used to socialise that much, only to see the family”. Furthermore, another said: “We can be pretty productive, and I don’t really feel as though we have suffered particularly”.

##### Aspects Changed

Participants described four areas they felt had changed in relation to their wellbeing and QoL as a result of the COVID-19 restrictions. The first of these related to their behavioural changes, as they had to adjust to adhere to social distancing guidelines. For this group of extremely clinically vulnerable individuals, this mandated complete isolation if possible, including from exercising outdoors. One individual noted: “Yes, well obviously, it affected what I do when I go out. I have been out a few times and if I have gone out at all it has been very early in the day when there are very few people about, and I do tend to give people a very large margin when I do go out walking”. Another participant described changes in communication methods saying: “It has changed, in the way that we normally speak and see people. We tend to have a lot of people coming around to see us for coffee, not parties or anything but just general, so that has been quite different, but we have done some stuff on Zoom and texting and stuff like that”.

A second area of change was described as feelings of independence and captured the emotional responses felt by participants through changes caused by COVID-19. When asked to describe how they felt, one participant expressed upset: “In the point that none of us can live our lives the way we would like to”. The third area of change was described as the emotional impact of the COVID-19 restrictions, where participants felt left behind or disappointed due to their vulnerability to COVID-19, with one participant saying “I was a bit disappointed to receive it [the letter informing them of their vulnerability status], but that is something which is just personal disappointment that I wish I wasn’t that much more vulnerable than anyone else”. Moreover, another participant stated that: “I was very tentative and didn’t really know what to expect and I felt like the rest of the world had already done it [left the house] and I was the last person to do it, if you like”. The final area of change was described as an increased concern, including concerns for job security, obtaining essential supplies and feeling safe to leave the house. When probed, participants expressed their “only worry was with work, was I going to have a job at the end of this isolation and the end of the shielding”, “That was really difficult to begin with, we really struggled with that [obtaining essential supplies] to begin with because we couldn’t get a supermarket slot” and “I would be concerned about going into hospital because I’ve heard a few stories of people going in and actually catching [COVID-19] it in hospital”.

#### 3.1.3. Dimension: Medical Care

##### Aspects That Have Remained the Same

When individuals described their medical care during the initial COVID-19 restrictions, two distinct unchanged experiences were noted. First, confidence in renal staff, with individuals describing their maintained confidence and outspoken praise for their clinical teams. For example, when asked about any changes to their medical care as a result of COVID-19 and the restrictions put in place, one participant said: “No, none at all, it has all been as it always has been. I am very applauding and grateful for that because at no stage was there any feelings of fear or anxiousness”.

The second unchanged experience was the support from the HHD team, with individuals describing a constant and proactive approach from the HHD team to ensure their welfare. For example, one participant said: “Well actually, a bit more support this time. Since the COVID-19 lockdown, I have had more support than before. I mean, before the support was always there available but now it is more proactive where I actually get called”.

##### Aspects That Have Changed

Participants described two areas where the COVID-19 restrictions have resulted in change in their medical care. The first of these related to communication difficulties, with some individuals describing incidents where communication was lacking, with one individual saying “Yes, I mean, to be honest with you, that has been the [most] frustrating thing of all, is not knowing what the results were and being able to actually talk them through”. One participant also described how the proactive response of the HHD team may have been unnecessary: “For a while after a few weeks ago, I was having weekly phone calls and I almost found them unnecessary, because I didn’t need them before, my situation hasn’t changed in the fact that I am a home patient so I knew that if I had an issue, there would be someone there for me”.

The second change related to concerns with future care where individuals expressed their concerns regarding future transplants and concerns returning to clinical environments: “Yeah, we are concerned that if it’s going to interrupt the prospect of transplants with the people and operations and all the rest of it so that concerns me”.

#### 3.1.4. Dimension: Experience of Telemedicine

Throughout these discussions, participants described two areas relating to the use of telemedicine during the COVID-19 restrictions. The first of these was support for telemedicine, where individuals expressed their support for the use of telemedicine and described positive experiences they have had utilising remote medicine within their clinical care: “Oh, I would be more than happy actually [to continue remote clinics]. It saves me travelling a long way. Absolutely fine, I would prefer [to continue remote clinics], I think”. However, individuals generally expressed their opinion that the use of video conferencing would be more beneficial than a telephone call, as it allows the use of nonverbal communication: “It would be better if you could do it via video link because then you can see the other person”. These experiences were for routine care where there is no particular issue, and many expressed that should an issue arise, they would rather a face-to-face consultation: “Yes, that would be fine to say yeah, I mean, obviously, if I did start not feeling well, or things, then I would like to see face-to-face but in general, I can’t see a problem with that”.

The second area relating to the use of telemedicine, was concerns about remote medicine, where individuals expressed their dislike for the formality of telephone appointments and the lost elements of nonverbal communication when utilising telemedicine: “I probably wouldn’t mind for the next three months when things are still settling or whatever, but I honestly don’t think you can beat hands-on seeing somebody”.

### 3.2. ICHD Group

#### 3.2.1. Dimension: PA

##### Changes in PA

Within the changes in PA, two areas were described. These two areas were changes in frequency and changes in modality. When this topic was discussed, most people fell within these areas. One individual discussed their reduction in frequency as “My PA was walking, and, in the house, there isn’t a lot of walking that you can do. They say to run up and down the stairs, but I can barely make it upstairs once let alone run up them”. Another described a change in modality as “Well there is a strange thing called housework and that has been receiving more attention than in the past, shall we say”.

#### 3.2.2. Dimension: Wellbeing and QoL

##### Aspects Maintained

During the discussions, all participants expressed two aspects of wellbeing and QoL that have remained the same during the COVID-19 pandemic. The first aspect is described as that of no change where many individuals described the lack of changes to their lives compared to the time before the pandemic. For example, two individuals said: “I can’t say that it has affected me a great deal” and “To be perfectly honest, it [COVID-19 restrictions] has changed our lives very little”. Secondly, many people reported that they were accepting of vulnerability and did not seem to be affected by their addition to the clinically extremely vulnerable list: “No, I don’t think it affected me. If something is going to happen, it is going to happen, isn’t it?”

##### Aspects Changed

When discussing aspects that have changed, five areas that have changed over this period were identified. The first of these being an impaired wellbeing, where individuals have described how the changes made during the COVID-19 pandemic have affected their lives: “It has upset my complete wellbeing. I was in a routine for so long and then all of a sudden it changed”.

A second aspect that had changed was social interactions, with all people interviewed who were receiving ICHD describing a reduction in their social interactions, but an uptake of social media use. One individual said “Oh, unbelievably so. I was talking to my partner, I would say 85% of the time and conversations get a bit strained after a while, there is only so much we can talk about”.

Issues with societal restrictions were also noted, with individuals describing how they were out of touch with social distancing and societal requirements due to their time shielding: “I went into the main door and I walked straight to where we normally go to speak to a waiter and he said sorry madam, back to the cross on the floor because I am not used to it”.

Concern for the future was also identified, with many describing their worry and anxiousness for what is to come: “A bit scared for the future quite honestly, because I don’t think that’s the last we are going to see of it. I have a feeling that once the winter sets in, it will be back”.

The final aspect is one of a positive experience where individuals have identified communal support whether this be from family, neighbours, friends or local councils to assist them throughout the pandemic: “The council here, have been very good. They have tried to send us a food parcel and all of that”.

#### 3.2.3. Dimension: Medical Care

##### Aspects That Have Remained the Same

When discussing their medical care, two themes were identified to have remained stable over this time. The first of these was administrative issues, which involved the difficulties with dialysis sessions, test results and facilitation of routine care: “Well obviously, they have moved me from QA [Queen Alexandra Hospital, Portsmouth] to Havant, and that was just a total nightmare for the first 4 to 6 weeks, it was horrendous” and another saying “Well, they come too late to get you to your appointment and they don’t take into account the fact that they bring you in late and they expect you to be ready on time”.

The second of these is communication difficulties with staff, where many issues and difficulties were caused by communication, or lack thereof with members of staff: “There has been no follow-up, no how are you getting on? No, this that or the other”.

##### Aspects That Have Changed

When discussing their medical care throughout the COVID-19 pandemic, three changes experienced were identified. The first of these is that some of individuals have reported to look forward to dialysis as it acts a break from the lockdown and the restrictions and provides social interactions: “But I mean as far as I am concerned, in terms of the pandemic, it was a bit of a release to be honest of you being able to get out of the house three times a week to go to the dialysis and see other people”.

The second of these is the confidence in renal staff, where individuals have described the confidence that they feel from the nursing staff taking care of them: “Yes. Yes, as I said, they are really good there, so they all talk to you and if you feel down or you want to talk to someone, they are there. You know, that we all have a laugh and that”.

The final experience identified is surrounding concerns for attending hospital, where many individuals have expressed their wariness of attending hospitals for routine appointments and even for dialysis: “Yes. I felt more at risk, obviously even though I was putting on PPE to go into a taxi, so I felt more at risk doing that and going to the dialysis centre than because of the environment”.

#### 3.2.4. Dimension: Experience of Telemedicine

When discussing the experience of telemedicine two themes were identified. The majority of the responses were in support for remote clinics, describing the benefit of reduced travel and waiting time and not having to attend the hospital for a quick chat with one describing it as “Providing the physicians are up to speed with what is going on in your life and your records, I think it is an extremely good way of doing it. It cuts down on travel time and their time”. Certain points were raised however in support of face-to-face appointments, when the patient or clinician identified a problem that they would like to discuss.

The second theme was regarding the dislike of remote clinics, where individuals described how they enjoyed the conversation and took comfort in seeing the clinician taking notes and providing an action plan in person. For example, when questioned, an individual said: “Yes, because I think if you talk to someone face-to-face about your problems and what you have got going on, I find that they understand and listen more. If you do it over the phone, I don’t know, I just feel like, I know they are good and they do take stuff on board but, you know, it’s just like a phone call, you know what I mean”.

## 4. Discussion

The primary aim of this study was to explore the impact of initial COVID-19 shielding restrictions to human movement (March–July 2020) on PA, wellbeing and QoL of ‘extremely clinically vulnerable’ adults with ESRD in the UK who were receiving ICHD or HHD. Furthermore, this study sought to ascertain their experiences of an increased reliance on telemedicine within clinical care. The principle findings were that, irrespective of dialysis modality (HHD or ICHD), PA, QoL and wellbeing, as well as medical care, were all impacted during this period. All individuals receiving ICHD reported negative consequences to their PA, QoL, wellbeing and medical care, whereas conversely, those receiving HHD primarily reported feelings of little change or a positive impact on their day-to-day life. However, a positive observation was the widespread support for the use of telemedicine within clinical care, regardless of whether people were attending ICHD or dialysing at home.

Early in the COVID-19 global pandemic it became apparent that, although shielding aimed to reduce the health risks posed to more vulnerable individuals, it also had the potential to bring about a number of negative physical and psychosocial health consequences. For example, a major immediate consequence of the restrictions to human movement was a reduced ability for many people to perform their typical frequency and preferred modality of PA [12], particularly in those advised to shield from the virus [6]. In the present study, all people receiving ICHD, with the exception of one individual who chose to act against recommendations, described a significant reduction in the frequency of their PA as a result of their shielding from COVID-19, as well as many of the HHD group.

Reduced PA as a result of the COVID-19 related restrictions is particularly concerning given that people with ESRD were generally engaging in less PA than recommended prior to these constraints, with a previous trial in the United States reporting that 56% of their cohort of people with ESRD were only active once or less per week [33]. This negative change in PA behaviour during the COVID-19 pandemic is, however, in line with other groups with long-term health conditions. For example, a recent investigation in people with cystic fibrosis reported lower PA in half of the participants surveyed during the COVID-19 lockdown in Switzerland with reasons for this including training facility closures, lack of motivation, and cancelled supervised training [34]. This is concerning, given the associated detrimental physical and psychological health effects of inactivity, which may well have greater implications in people with chronic health conditions. Indeed, only a few days of inactivity can induce skeletal muscle loss, reduce aerobic fitness (an important health marker) and worsen insulin resistance [35]. Specifically, in adults with well-controlled type 2 diabetes mellitus, the COVID-19 lockdown induced metabolic worsening in approximately a quarter of their sample [36,37]. Importantly, type 2 diabetes mellitus is a frequent comorbidity in people with renal failure [38,39], with an estimated 30% of people with diabetes in the UK having CKD stage 3–5 [40] and is the most common cause of ESRD in most countries [41].

Prior to the COVID-19 pandemic, wellbeing and QoL were often reduced in people with ESRD [42,43], especially when receiving dialysis [44]. Since physical inactivity was already associated with greater levels of anxiety and depression in people with ESRD receiving ICHD pre-COVID-19 [21], it is conceivable that this may have been further impacted whilst shielding from the virus. Indeed, a recent multicountry analysis showed a negative change in PA behaviour during initial lockdown restrictions to be associated with poorer mental health and wellbeing in the general population [12], with additional UK-based data reporting higher levels of PA during this time to be associated with better mental health [45,46]. The association between PA and wellbeing and QoL is well documented, with many reporting better health-related QoL and perceptions of wellbeing in people who are more physically active [33,47,48,49]. A recent study in 18–78 year olds during nationwide lockdown restrictions in Spain noted increased levels of anxiety and depression during the COVID-19 pandemic [11]. Furthermore, greater levels of anxiety have also been reported in people with cancer during the COVID-19 restrictions [50,51], as well as individuals with rheumatological conditions reporting a worsening of their health-related QoL [52].

In the present study, it was not only the volume but also the type of PA that was impacted in adults with ESRD, with all interviewed participants from both groups reporting that this had changed. Those who has previously undertaken frequent trail and group walking, cycling and personal training type activities reported a shift to walking alone, indoors and outdoors, and an increase in home-based activities, such as housework and/or gardening. In contrast to recent findings in people with cystic fibrosis [34] and general population data [12], only one of the participants in the present study reported using social media or digital health technology to help them be physically active during this period. Interestingly, several participants did express a desire to participate in this type of activity, however, they stated that they did not know where to begin.

Although general recommendations regarding alternative home-based PA during COVID-19 restrictions have been shared for the general population [53,54], and various generic online exercise platforms are available, the present findings highlight the need for safe methods which are more specifically tailored to allow people with renal disease to exercise at home [55]. Promisingly, following these interviews, Kidney Beam (https://beamfeelgood.com/kidney-disease) has been developed; an online platform which aims to help with PA and emotional wellbeing in people with kidney disease that is free for people in the UK until 30 November 2021.

In addition to PA, other contributors to reduced wellbeing and QoL were reported in our cohort of shielding adults with ESRD. Many described reduced social interactions, with this being particularly apparent in people receiving HHD compared to those attending for ICHD. Although attending dialysis has been highlighted as an increased risk factor for contracting COVID-19, partly due to the large groupings of people within a relatively small clinical area [3], our cohort of people receiving ICHD during this time actually expressed an eagerness to attend their dialysis sessions as it provided them with a break from their enforced isolation and, with this, an opportunity to speak to other people. This was highlighted as being particularly important with individuals from both of our study groups reporting increasingly stressed familial relationships during this time. These findings support recent evidence from Sousa et al. [56], who documented increased stress and anxiety surrounding the COVID-19 virus in an ICHD population in Portugal, with people often suffering in isolation. Similar experiences were captured in the present study, with many reporting feeling that they had lost their freedom and independence. This was particularly so in people receiving HHD, likely due to their previously increased independence.

The secondary focus of this study was to capture the experiences of adults receiving ICHD and HHD with regards to changes to their clinical care and the necessary increased reliance on telemedicine. More specifically, there was a need to move routine medical care to virtual delivery and for some ICHD dialysis sessions to change location. A primary positive finding was that people receiving HHD during this time period felt that they received constant proactive support, in the form of consistent and regular phone calls. Since the treatment of people receiving HHD is largely independent and requires minimal day-to-day input from clinicians, a largely unchanged treatment experience was noted in this group, despite reduced face-to-face contact with their renal clinician. People dialysing at home also reported few issues or concerns with regards to their medical care. However, the experience of people receiving ICHD was affected to a greater extent.

The clinical care of people receiving ICHD requires input from dialysis nurses and therefore incorporates travelling to a clinical facility three times per week. The ICHD group in the present study reported a number of administrative issues and communication difficulties, which at times resulted in problems with their medical care, such as delayed prescriptions, missed clinic appointments and even missed dialysis sessions. Similar issues to our cohort in the UK were identified in adults receiving ICHD during a COVID-19 lockdown in India where specifically, COVID-19 had similar effects on the dialysis infrastructure resulting in missed dialysis sessions and appointments and, consequently, a negative patient experience [57]. These observations within renal disease are in line with a global-survey in health care professionals caring for people with a variety of chronic diseases using adapted delivery methods during COVID-19, in which 67% perceived moderate-to-severe negative effects on their patients [58].

A positive finding in the present study was, however, the experiences with telemedicine and an increased use of digital health technologies, which lends support to the NHS five-year forward view to use more technology within routine care [59]. More recently, the Carter report [60] highlighted the need to look at digital solutions as a way to promote efficiency within the NHS. Those who dialyse at home within the Wessex Kidney Centre were already familiar with MyRenalCare^®^ and were more accustomed to remote medicine and feedback than those dialysing in-centre. Generally, individuals in this study supported the use of remote clinics and telemedicine saying that it was nice to reduce waiting and commuting time. This supports recent findings [61], which found a high level of satisfaction with telemedicine when used in an Ear, Nose and Throat (ENT) department throughout the COVID-19 restrictions.

Prior to COVID-19, the use of telemedicine to perform renal clinics was shown to be accepted by clinicians with benefits of reduced travel time, more efficient use of staff time, and a strong sense of job satisfaction being reported [62], akin to other clinical specialties [63,64,65]. A recent publication by the American Society of Nephrology outlined the expectation for telemedicine to remain within renal medicine in some form following the COVID-19 pandemic and associated restrictions [66]. Moreover, a similar study conducted during COVID-19 in renal clinicians also reported widespread support for telemedicine, reporting more efficient use of staff time, reduced travel, peace of mind and a strong sense of job satisfaction [67]. The initial experience of the participants in this study is therefore valuable to help inform what this future clinical management may look like.

Several participants also had negative experiences with the current use of telemedicine or aspects of it, with some highlighting that they felt that the face-to-face interaction with their consultant or medical professional would provide more comfort in the depth of care they were receiving. This is in line with findings by Fieux et al. [61], which found that clinicians acknowledged a lack of complete medical care caused by reductions in hospital outpatient procedures, within an ENT department. Some of the alternatives suggested by participants in the present study included video conferencing rather than telephone calls, using remote clinics every other appointment or the use of remote clinics when there is no concern, with the option of face-to-face when the patient had something to discuss, as well as an opt-in mechanism for remote care. The findings from this study lend support to the use of remote clinics within the clinical management of people with ESRD receiving HHD or ICHD in the future, however, lessons can be learnt from delivery during the current pandemic and enhancements made [68,69]. This study also provides initial insight into patient feedback of using the MyRenalCare^®^ application which, although originally developed for HHD, may have potential use for the monitoring of people receiving ICHD, particularly on nondialysis days. Moreover, this study highlights the need for more proactive care if/when people may be asked to shield again and support for the implementation of telemedicine in routine care, in line with the NHS forward view [59,60] and the NHS Long Term Plan [28]. Finally, the recently developed Kidney Beam platform is an example of the implications of digital platforms to promote PA, with this study supporting the use of telemedicine to promote independence in patient care.

The present findings need to be interpreted in the context of several methodological considerations. First, it has been acknowledged that the modality may alter the interview and, more specifically, the use of telephone interviews for qualitative research is considered by some to reduce the quality of data obtained [70]. However, this method has been used successfully in previous research both during the current pandemic and in previous research into QoL of adults with ESRD [71,72]. Nonetheless, the choice of telephones, as opposed to online video platforms, was informed by patient and public involvement during the trial design. It would, however, be of interest to follow up some of these individuals with face-to-face interviews when social restrictions are not in place. Second, whilst the use of semistructured interviews enabled the discussion of a broad range of issues, including PA, it is appreciated that PA was not objectively measured. Therefore, further research to investigate the PA behaviours of this population using validated accelerometers as we progress through the COVID-19 pandemic and ongoing restrictions to human movement is warranted. Finally, longitudinal follow-up of these individuals is important to understand any longer-term impact of the COVID-19 pandemic on their PA behaviours, QoL and wellbeing, as well as identifying any need for additional rehabilitation.

## 5. Conclusions

In conclusion, this study provides valuable insight into the experiences of shielding adults with ESRD, receiving ICHD and HHD in the UK, during the initial COVID-19 restrictions. During this period, PA behaviours, wellbeing and QoL, as well as confidence in care in the ICHD group, were all negatively affected. The majority of people interviewed had positive experiences of the increased use of telemedicine within their clinical care. These findings highlight the need for more proactive care if people are asked to shield again, as well as increased awareness of safe and appropriate PA resources to help with home-based PA and emotional wellbeing, such as the newly developed Kidney Beam platform.

## Figures and Tables

**Figure 1 ijerph-18-03144-f001:**
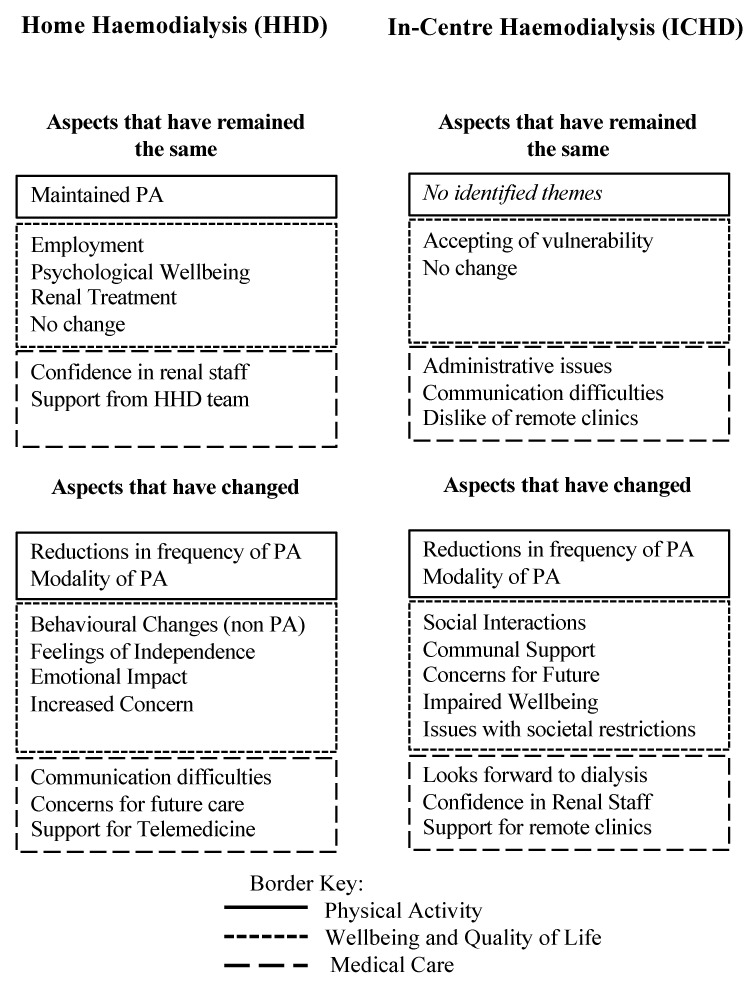
Summary of the themes identified during thematic analysis. These denote the key thematic areas discussed as having been changed or not changed as a result of the COVID-19 restrictions. N.B. PA, physical activity.

## Data Availability

The data presented in this study are available on request from the corresponding author.

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
