# Peer review of "Understanding the Impact of Initial COVID-19 Restrictions on Physical Activity, Wellbeing and Quality of Life in Shielding Adults with End-Stage Renal Disease in the United Kingdom Dialysing at Home versus In-Centre and Their Experiences with Telemedicine"

_ijerph, 2021, doi:10.3390/ijerph18063144_

Round 1
Reviewer 1 Report
Telemedicine is an interesting topic. Just few points:
- Lines 17-20 "The aim of this study was to investigate how these restrictions...haemodialysis (ICHD) in the UK." At the end of Introduction section it is better to remark what is the aim to help the readers in the thier reading.
- Lines 165-168. The study was conducted on 20 patients. Please add this data in the result section at this point.
- Lines 63-70. The authors reported clinical practice common changes occurred during this pandemic. I think the meaning of this paper is to pay attention to Covid-19 but not to forget other pathologies like chronic kidney disease (CKD). Please include this paper about Covid-19 topic at this point: Intracranial hemorrhage and COVID-19, but please do not forget "old diseases" and elective surgery. Brain Behav Immun. 2020 Nov 25:S0889-1591(20)32410-7. doi: 10.1016/j.bbi.2020.11.034.
- Lines 390-401. The role of telemedicine during COVID must be highlighted more. Please add this two very important references at this point: Montemurro N & Perrini P. Will COVID-19 change neurosurgical clinical practice? Br J Neurosurg. 2020 Jun 1:1-2. doi: 10.1080/02688697.2020.1773399. Ganapathy K. Telemedicine and Neurological Practice in the COVID-19 Era. Neurol India. 2020 May-Jun;68(3):555-559. doi: 10.4103/0028-3886.288994.
- Lines 559-560. I think that the paper has some limitations. Please add them at this point.
- Figure 1 is good idea, even if it is not easy to read. Maybe you can add some more words in the legend to explain it better.
Reviewer 2 Report
I would like to commend the authors for their work. There are, however, some minor clarifications needed:
- report ethical approval by the committee;
- report how wellbeing and QoL was measured;
- propose practical implications on how health professionals can endorse more digital/telemedicine to increase wellbeing, QoL, and PA among ERSD patients.
Reviewer 3 Report
Well-written manuscript; however there are few areas where I have recommendations:
Page 1 line 41: don't think you need the exact date just having AprilPage 2 2020 is good
Page 2 line 89: did the authors receive IRB approval
Page3
Line 95: how did the authors determine psychiatric or neurological diagnoses? did they exclude all (e.g., depression, stroke with no residual affect, neuropathy)
line 101: would not say blood purification because that is not what HD does
Page 7, line 250: how did being hospitalized for COVID impact their health
In the discussion it is important to discuss next steps: research, practice, policy? what were the limitations of the study and recommendations of how to address in next study
Round 2
Reviewer 1 Report
Authors solved all criticisms.